# Molecular Profiling in Non-Squamous Non-Small Cell Lung Carcinoma: Towards a Switch to Next-Generation Sequencing Reflex Testing

**DOI:** 10.3390/jpm12101684

**Published:** 2022-10-09

**Authors:** Nina Pujol, Simon Heeke, Christophe Bontoux, Jacques Boutros, Marius Ilié, Véronique Hofman, Charles-Hugo Marquette, Paul Hofman, Jonathan Benzaquen

**Affiliations:** 1Centre Antoine-Lacassagne, Department of Radiation Oncology, Côte d’Azur University, 06000 Nice, France; 2Department of Thoracic/Head and Neck Medical Oncology, MD Anderson Cancer Center, Houston, TX 77030, USA; 3Laboratory of Clinical and Experimental Pathology, Côte d’Azur University, Pasteur 1 Hospital, Centre Hospitalier Universitaire de Nice, FHU OncoAge, Biobank BB-0033-00025, 06000 Nice, France; 4CNRS UMR 7284, INSERM U1081, Institute of Research on Cancer and Aging, Côte d’Azur University, 06000 Nice, France; 5Department of Pulmonary Medicine and Thoracic Oncology, Côte d’Azur University, Pasteur 1 Hospital, Centre Hospitalier Universitaire de Nice, FHU OncoAge, 06000 Nice, France

**Keywords:** next-generation sequencing, predictive biomarker, non-small cell lung carcinoma, precision oncology

## Abstract

Molecular diagnosis of lung cancer is a constantly evolving field thanks to major advances in precision oncology. The wide range of actionable molecular alterations in non-squamous non-small cell lung carcinoma (NS-NSCLC) and the multiplicity of mechanisms of resistance to treatment resulted in the need for repeated testing to establish an accurate molecular diagnosis, as well as to track disease evolution over time. While assessing the increasing complexity of the molecular composition of tumors at baseline, as well as over time, has become increasingly challenging, the emergence and implementation of next-generation sequencing (NGS) testing has extensively facilitated molecular profiling in NS-NSCLC. In this review, we discuss recent developments in the molecular profiling of NS-NSCLC and how NGS addresses current needs, as well as how it can be implemented to address future challenges in the management of NS-NSCLC.

## 1. Introduction

Lung cancer is the leading cause of cancer-related death, as well as the third most frequent cancer in Europe with an increasing incidence [1]. Non-small cell lung cancer (NSCLC) is the most frequent lung cancer, with two predominant histologies, non-squamous (NS) and squamous cell carcinoma (SCC). Smoking remains the leading risk factor for non-small cell lung cancer, though cases related to low or no smoking burden are increasingly clinically relevant [2,3]. Importantly, while targetable alterations are more commonly found in patients with lower smoking burden, targetable molecular alterations must be assessed independently of smoking status for any patient with NS-NSCLC, as these alterations are found in ~50% of cases in non-smokers and up to 20% of cases occurring in former or current-smoking patients [4].

Personalized medicine revolutionized NS-NSCLC in 2004, with the discovery that specific mutations in the epidermal growth factor receptor (EGFR) sensitized tumors to a targeted treatment with the tyrosine kinase inhibitors (TKI) erlotinib and gefitinib [5,6,7]. Following approvals in a later treatment setting, the first-line administration of EGFR TKIs became standard in 2009 with the approval of gefitinib [8], in patients with NS-NSCLC harboring an *EGFR* deletion 19 or L858R substitution, followed by the approval of erlotinib [9] and afatinib [5].

Despite durable responses with the administration of EGFR-TKIs, resistance systematically occurs, and the occurrence of the *EGFR* T790M resistance mutation has been described as a major driver of resistance [10]. This led to the development and approval of third-generation EGFR TKIs like osimertinib, which are currently recommended for first-line use in patients with *EGFR* mutant NS-NSCLC with exon 19 deletion or L858R mutation [11]. However, resistance remains a major challenge in this setting, with increasingly complex and diverse resistance mechanisms [12]. As an example, C797S mutations, which confer resistance to osimertinib treatment, are currently being targeted with the development of fourth-generation TKIs [13,14].

Independently of *EGFR*, multiple additional drivers have been defined in NS-NSCLC, and many drugs have been approved in the last few years that directly target those alterations. Anaplastic lymphoma kinase (*ALK*) and *ROS-1* proto-oncogene rearrangement, *BRAF* proto-oncogene mutation, V-Ki-ras2 Kirsten rat sarcoma viral oncogene homolog (*KRAS*) mutations, neurotrophic tyrosine receptor kinase (*NTRK*) fusion, *MET* proto-oncogene mutation or amplification, *RET* proto-oncogene mutation or fusion, Neuregulin 1 (*NRG1*) fusion, and Human Epidermal Growth Factor Receptor-2 (*HER-2*) mutation are all described in NS-NSCLC, with corresponding drugs investigated or approved.

Those developments reshape the classification of lung cancer based on molecular classification and enabled an increasingly complex personalized approach in thoracic oncology, which improved the overall survival but also the quality of life of these patients [15]. 

Consequently, tumor genetic testing has become standard of care for metastatic and locally advanced NS-NSCLC [4]. Indeed, the Food and Drug Administration (FDA) and the European Medical Agency (EMA) promote the principles of precision medicine and claim that a specific and approved companion biomarker is mandatory for the safe administration of a targeted drug [16]. *EGFR*, *ALK*, *KRAS*, and *ROS1* testing are mandatory at initial diagnosis, while *BRAF* mutations, *MET* exon 14 mutations, *RET* fusion, and *NTRK* fusion testing are optional at initial diagnosis and mandatory before second-line therapy, and other *MET* mutations, *HER2* mutations, and *NRG1* fusions should be tested before second-line treatment [17]. For example, second-line trastuzumab–deruxtecan demonstrated clinical benefit in metastatic *HER2*-positive patients, with a recent FDA approval for this indication highlighting the constantly evolving need for biomarker testing [18]. In addition, *EGFR*-mutation testing for pII and pIII stage NS-NSCLC patients is anticipated to become mandatory based on the results of the ADAURA study positioning the use of osimertinib as adjuvant therapy in this population [19].

After an initial response, drug resistance and disease progression occur in most cases, and therefore companion biomarkers must not only identify a therapeutic target at diagnosis, but also the techniques must be repeated, as therapeutic resistance mechanisms should be investigated during disease progression. Several methods are used to identify these molecular alterations, by analyzing protein expression levels using immunohistochemistry (IHC), by assessing mutations in tumor DNA, and by assessing the fusion of certain genes of interest (i.e., *ALK*, *ROS1*, *NTRK*, *RET*) using IHC and notably fluorescence in situ hybridization (FISH) [15]. A reference method for the study of mutations in tumor DNA was historically Sanger sequencing, described in 1977, which has been superseded by the routine use of real-time PCR (qPCR) [20]. A still commonly used strategy is based on sequential exclusion testing looking for mutations that are mutually exclusive with others or using hotspot screening to search for exploitable molecular alterations. In contrast, next-generation sequencing (NGS) allows for broad molecular profiling and allows for the simultaneous detection of both common targetable and rare mutations at the tumor DNA level and are able to identify translocations at the RNA level [20]. 

Practically, *EGFR* including the T790M mutation, *BRAF* and *KRAS* mutations, are commonly analyzed by qPCR; ALK rearrangement can be analyzed either by IHC or fluorescence in situ hybridization (FISH); ROS1 rearrangements relies on IHC and has to be confirmed by FISH analysis, and more recently DNA and RNA NGS are used for a larger panel of molecular alterations including *MET* exon 14; *RET; NTRK1,2,3*; and *NRG1* [17]. In addition to *MET* exon 14 skipping assessment by NGS, the evaluation of the *MET* gene copy number (amplification) by FISH (gold-standard) or NGS should be an integral part of the diagnostic work-up, due to its clinical relevance [21]. Indeed, *MET* amplifications may occur de novo or as a mechanism of resistance to EGFR TKIs, but clinical trials have demonstrated that this could be targeted using capmatinib [22] or tepotinib [23] as second-line therapy [24]. 

## 2. Approved Companion Tests (Figure 1)

### 2.1. EGFR

The *EGFR* gene codes for a membrane receptor with tyrosine kinase activity belonging to the *HER* family [25]. The activation of EGFR is dependent on ligand binding to the extracellular domain resulting in a conformational change of the receptor allowing its hetero or homo-dimerization [26], leading to the phosphorylation of the tyrosine kinase domain. This activation allows various downstream signaling pathways of cell proliferation. The two main types of *EGFR* mutations are represented by in-frame exon 19 deletions, which are the most frequent, and L858R exon 21 substitution [27], and are targetable by first (erlotinib, gefitinib)-, second (afatinib, dacomitinib)-, and third (osimertinib)-generation TKIs now approved for first-line treatment in the European Union [28]. Afatinib is additionally approved for treating NSCLC with G719X, S768I, and L861Q mutations. 

**Figure 1 jpm-12-01684-f001:**
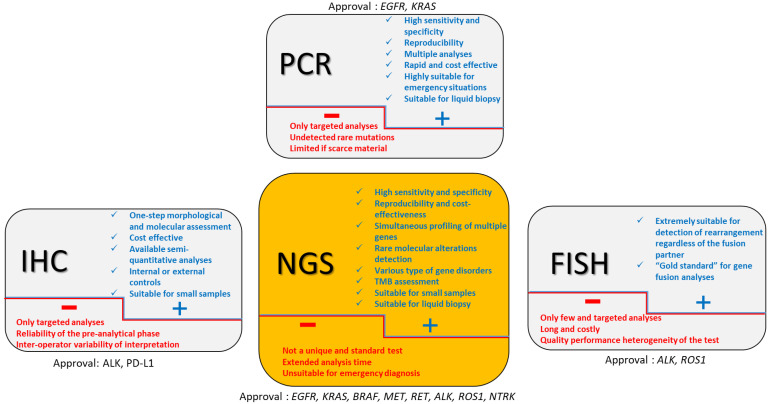
Approved diagnostic tools in lung cancer. Strengths and weaknesses of diagnostic tools appear respectively in blue and red font.

qPCR is the most routinely used tool to detect deletion 19, exon 20 mutations, exon 21 L858R substitution, and T790M mutation and allows for a particularly rapid result [29]. However, some allele-specific PCR kits are likely to miss some rare *EGFR* mutations like exon 18 alterations (E709X, E709A, E709G, E709K, E709V, delE709-T710insD, G719X, G719S, G719A, G719C, and G719D), exon 19 in-frame insertions, exon 20 alterations (A763_Y764insFQEA insertion, S768I), and exon 21 mutation L861Q [30,31,32].

In order to use EGFR TKI, diagnosis by qPCR based on the Cobas^®^ or Therascreen^®^ assay is approved for the detection of EGFR mutations and used routinely, but only Cobas is approved for the use of osimertinib in patients with *EGFR*-mutated non-small cell lung cancer (NSCLC) with T790M mutations, and only Therascreen^®^ assay is approved for the use of dacomitinib. Of note, Therascreen^®^ assay is approved for using afatinib in L861Q, G719X and S768I mutations. 

Next-generation sequencing (NGS), either using tissue or circulating tumor DNA (ctDNA) [33] based on the FoundationOne CDX assay (Foundation Medicine, Cambridge, MA, USA)^®^ and FoundationOne Liquid CDX^®^ assay for ctDNA EGFR-mutations assessment and Guardant 360 (Guardant Health, Redwood City, CA, USA)^®^, are approved for the use of TKIs targeting exon 19 deletions and L858R exon 21 substitution (gefitinib, erlotinib, afatinib, osimertinib). Other approved tests are the ONCO/Reveal Dx Lung & Colon Cancer Assay (O/RDx-LCCA) (Pillar Biosciences, Inc., Natick, MA, USA)^®^ for gefitinib, erlotinib, afatinib, and dacomitinib and the Oncomine Dx Target Test (ThermoFisher Scientific, Waltham, MA, USA)^®^ for gefitinib, but also the promising new drugs amivantanab and mobocertinib, initially developed to target EGFR exon 20 insertion at second-line treatment and now investigated in other mutations as well [34]. Amivantamab is studied in the CHRYSALIS Phase I Study [35], with or without association with lazertinib in the CHRYSALIS-2 Phase Ib/II Study [36], and in association with different regiments of chemotherapy in the PAPILLON Phase III Study [37]; while mobocertinib has been studied [38] for targeting Exon 20 EGFR insertions, amivantamab recently obtained approval by the FDA and EMA, while mobocertinib was approved by the FDA. 

### 2.2. ALK

The *ALK* gene is located on chromosome 2 and codes for a receptor tyrosine kinase and was first identified in anaplastic large cell lymphoma [39]. Although many variants have been described with varying frequencies, the rearrangement of the *ALK* gene with *EML4* (Echinoderm microtubule associated protein like 4) is the most common mechanism and occurs due to a rearrangement on the same chromosome [40,41]. This rearrangement leads to the expression of a chimeric protein, with the constitutive activation of ALK leading to cell proliferation [40], and can be targeted by the ALK TKIs crizotinib and ceritinib or more recently alectinib, brigatinib, and lorlatinib [42]. 

The first studies evaluating *ALK* rearrangement were based on FISH; however, IHC is increasingly used. Nevertheless, confirmation is recommended using FISH (if available) to confirm the gene fusion event in case of a positive IHC [43].

ALK antibodies include ALK1, 5A4, D5F3, and SP8 with better diagnostic performance in terms of sensitivity and specificity for 5A4 (Novocastra^®^) and D5F3 (Ventana^®^) [44,45], resulting in their approval as companion diagnostics for the Ventana ALK (D5F3) CDx^®^ IHC Assay. FoundationOne CDx and FoundationOne Liquid (Foundation Medicine, Cambridge, MA, USA)^®^ are approved [46] for administrating ALK TKIs. NGS from RNA as an alternative to DNA-based sequencing has also been investigated and implemented [47] and may be critical for the comprehensive detection of *ALK* gene-specific fusion partners. 

### 2.3. ROS1

ROS1 is a tyrosine kinase receptor of the insulin receptor family that activates the MAPK signaling pathway through the phosphorylation of RAS. Its hyperactivation leads to cell growth and proliferation [48], and this receptor can be successfully targeted by crizotinib and entrectinib [49].

There are four approaches described to detect *ROS1* rearrangements: IHC, FISH, qPCR, and next-generation sequencing (NGS) [50,51]. IHC can be used as screening tool and needs confirmation by another method, either FISH, qPCR, or NGS [50,51,52,53]. Nevertheless, due to the low frequency of this alteration, *ROS1* rearrangements are not always routinely tested at baseline when NGS is not used and thus are assessed only after eliminating an alternative molecular alteration. Consequently, samples may have been exhausted by previous tests, exposing the risk of insufficient material, particularly for IHC or FISH techniques [54].

Consequently, two NGS companion tests are approved, either Oncomine Dx Target Test (ThermoFisher Scientific, Waltham, MA, USA)^®^ [55] or FoundationOne CDx (Foundation Medicine, Cambridge, MA, USA)^®^ [56]. 

### 2.4. BRAF

The *BRAF* gene encodes an intra-cytoplasmic serine/threonine kinase. *BRAF* mutations, mostly non-V600E, account for 1–4% of NS-NSCLC [57] and lead to the activation of the MAPK pathway downstream of EGFR and RAS, and this kinase can be targeted by the combination of the BRAF inhibitor dabrafenib and the MEK inhibitor trametimib [58,59].

The use of IHC is a promising approach; however, the antibody available for BRAF (VE1) [60] limits the detection to only exon 15 V600E mutations, while other alterations can be found [61,62,63].

Oncomine Dx Target Test (ThermoFisher Scientific, Waltham, MA, USA)^®^ and FoundationOne CDx (Foundation Medicine, Cambridge, MA, USA)^®^ are approved NGS companion tests for using dabrafenib–trametinib.

### 2.5. KRAS

The RAS protein is located downstream of the EGFR signaling cascade and can also interact with other EGFR downstream pathways (MAPK, PI3K-AKT-mTOR), thus bypassing its activation. This protein has an important role in cell growth, differentiation, and apoptosis control and is active when bound to GTP. *KRAS* is the most commonly mutated oncogene in NSCLC, and this mutation is associated with a poor prognosis [64], with the G12C being the most frequent (40–50% of *KRAS* mutations), followed by G12V (19% of *KRAS* mutations) [65,66]. These mutations are particularly observed in the lung adenocarcinoma (25 to 40%) of patients with high smoking exposure. While long considered to be undruggable, recently the approval of sotorasib revolutionized the targeting of *KRAS* mutant NSCLC, with adagrasib as another G12C targeting TKI expected to receive FDA approval following promising results from the phase II Krystal-1 trial as well [67]. Furthermore, there are many additional drugs targeting other non-G12C mutants currently under development [68].

The two companion tests approved are the Therascreen KRAS RGQ PCR Kit (Qiagen, Manchester, Ltd., UK)^®^ qPCR test and the ctDNA Guardant360 CDx (Guardant Health, Inc., Redwood City, CA, USA)^®^ NGS test for using sotorasib [69].

### 2.6. MET

The *MET* gene is located on chromosome 7 and codes for a receptor with tyrosine kinase activity whose ligand is hepatocyte growth factor. This gene is involved in the carcinogenesis of NSCLC, with several mechanisms leading to its activation by gene amplification in 4% of cases, activating the mutation and deletions of exon 14 and the overexpression of hepatocyte growth factor [70]. The occurrence of *MET* amplification increases to ~20% in 3rd TKI resistant *EGFR*-mutated NSCLC, constituting an important mechanism of resistance [70]. Presently, *MET* exon 14 skipping can successfully be targeted using tepotinib [71] and capmatinib [22].

FoundationOne CDx (Foundation Medicine, Cambridge, MA, USA)^®^ and FoundationOne Liquid CDx^®^ are two approved NGS companion tests for this alteration for using capmatinib, but there is currently no companion test specifically approved for using tepotinib.

However, an exclusive *MET* determination based on NGS cannot completely replace an assessment of *MET* gene amplification status based on FISH [21], which allows the precise assessment of amplification level normalized to the level of concomitant polysomy, although neither *MET* amplification test is approved.

### 2.7. RET

The *RET* gene translates into a proto-oncogene transmembrane receptor tyrosine kinase, leading to the activation of downstream signaling pathways such as RAS, MAPK, and PI3K-AKT-mTOR. The juxtaposition of the C-terminal region of the RET protein with the N-terminal portion of another protein by fusion, of which KIF5B is the most common fusion partner [72], leads to its self-sustained constitutive activation [73] and can be successfully targeted using pralsetinib [74] or selpercatinib [75].

The only approved companion test is the Oncomine Dx Target Test (ThermoFisher Scientific, Waltham, MA, USA)^®^ NGS test for using pralsetinib, whereas selpercatinib does not yet ahve a companion diagnostic test to date.

### 2.8. NTRK

The *NTRK* 1, 2, and 3 genes allow the production of three proteins, TRKA, TRKB and TRKC, which affect cell differentiation, survival and proliferation, and migration. The NTRK fusion protein results from a DNA fragment insertion into another part of the genome. This fusion protein has emerged as a target, with approved treatments by the EMA and the FDA for larotrectinib [76] and entrectinib [77], by accelerated approval for all adult solid tumors harboring an NTRK fusion. 

The only approved companion test is FoundationOne CDx (Foundation Medicine, Cambridge, MA, USA)^®^ NGS companion test, for using larotrectinib or entrectinib [78].

## 3. From Multiple Companion Biomarkers to NGS?

Many companion tests exist for NS-NSCLC therapeutic targets, though not all are being approved. However, the initiation of a targeted therapy is feasible even if the test used is not the native companion test, as shown in [79] where the switching of PCR assays for common *EGFR*-activating mutations did not affect the outcome.

Considering that most oncogenic driver mutations that can be targeted are mutually exclusive, sequential testing, in which each gene is tested in a stepwise approach, has long been the standard procedure in the molecular pathology of NSCLC [52].

However, this approach is increasingly challenged by the development of novel therapies, dramatically broadening the number of genes to be tested [80]. In NSCLC, where tissue is often limited, all available specimens will be exhausted before tissue testing is complete. Furthermore, this approach often leads to incomplete biomarker testing, limiting the possibility of initiating adequate treatment [81,82].

Additionally, many tests do not cover all relevant regions in the respective genes and thus often miss out on the detection of certain mutations. For example, there are several studies currently investigating the treatment of Exon 20 mutations in both *EGFR* and *HER2*, with multiple drugs currently being investigated in this setting [83]. Most qPCR tests only have limited coverage in those regions, thus limiting the detection of mutations and, consequently, the initiation of targeted treatments in this setting [84,85].

A recent study investigating the structural implications of a broad range of *EGFR* mutations highlighted that many *EGFR* mutations result in comparable structural changes and comparable sensitivity to certain TKI drug classes [31]. Since many of these mutations cannot be investigated with qPCR, the implementation of NGS is critical in the clinical setting. 

Indeed, the use of NGS as a primary test has dramatically increased during past years [86], though additional efforts are critical to implementing NGS widely as a principal testing method at baseline in NSCLC. This approach is not only suitable for primary tumor tissue but is routinely used in the transbronchial needle aspiration of lymph node [87], plasma [88], and pleural effusions [89] and could also be used from bronchoalveolar lavage fluid [90] and is also suitable for cerebrospinal fluid [91]. Besides, there are promising data for the screening of residual disease of *EGFR* mutant NSCLC in saliva [92] and urine [93].

The rationale to extend the use of NGS for NSCLC molecular diagnostic is also highlighted by ESCAT (ESMO Scale for Clinical Actionability of Molecular Targets). This collaborative tool aims to classify the molecular alterations that can be found in tumor tissue [94] or in ctDNA [95] into six tiers, depending on the clinical significance of the molecular alterations, in order to harmonize practices. Briefly, tier I corresponds to alterations that are known to be targetable; tier II corresponds to alterations that are non-targetable according to current recommendations but are promising in clinical trials; tier III corresponds to alterations that are targetable in another tumor type or for similar targets; tier IV corresponds to alterations for which preclinical models are promising; tier V corresponds to alterations that could be co-targeted, and tier X corresponds to alterations with no evidence of clinical significance. Importantly, studies have provided evidence that the ESCAT classification of genomic alterations is implementable in clinical practice [96,97].

In the past, especially increased costs and longer turn-around times have disfavored NGS over sequential testing utilizing IHC/FISH or qPCR. Comparisons on the cost efficacy of NGS are challenging due to a rapidly changing field and due to the fact that the assessment of genetic tests has traditionally been regulated mostly at the national level in the EU [98], leading to a certain heterogeneity of the pathology workflow between countries and institutions. In France, for example, regional molecular genetics centers have been created to allow performing selected molecular tests free of charge for all cancer patients in their region, with financial support assisting in the implementation of NGS [15]. Issues surrounding the systematic implementation of NGS at the European level are discussed in [99], highlighting difficulties linked to the governance of collaboration between private and public institutions, to the adaptation of the necessary and often complex infrastructure, to the standardization of sequencing panels which might include in-house developed assays, to the often diverse training of personnel, and to the financial cost of the tests. Importantly, cost effectiveness was investigated in multiple studies, and NGS has been considered cost-effective in NSCLC when multiple genes require parallel assessment [100,101,102]. However, there are still ongoing debates on the most cost-effective use of NGS. For example, while performing NGS at initial diagnosis as well as at disease progression under treatment is technically feasible and routinely performed, reimbursement for NGS in France is limited to one single test. Multiple testing is thus often covered by research funding or results in additional out-of-pocket costs for the patients. Similar regulations may exist in other countries as well, and consequently reimbursement regulations need adaptions in order to allow a scientifically driven and effective biomarker-driven treatment in NSCLC. 

Two approaches can be currently considered for NGS [103]. The “bespoke testing” approach, initiated “on-demand” by the clinician, is essentially aimed at advanced-stage patients in order to initiate a targeted therapy. However, this approach might delay treatment duration or might restrict the testing to certain population, due to delay in the request for testing and the tissue exhaustion due to multiple requests at different times. By contrast, the “reflex testing” approach is triggered by a pathologist based on a histological diagnosis, with the advantage of managing currently available tissue more efficiently, with an early result, which, however, leads to increased testing requirements and requires additional resources and infrastructure. 

In addition to tissue-based testing approaches, liquid biopsies using circulating tumor DNA (ctDNA) are also critical for clinical recommendations and are thus recommended by the IASLC [104]. However, the importance of liquid biopsies compared to tissue have been reviewed extensively, and the following paragraph should only give a brief overview [105,106,107,108]. Importantly, liquid biopsies can help to adjust treatment decisions under active treatment through the detection of resistance mechanisms but can also replace tissue-based testing when specimen have been exhausted or are too limited [109], a situation in which the validation of NGS assays designated for low DNA input is crucial. It can further help as a prognostic tool through the determination of minimal residual disease (MRD) [107]. Consequently, it is critical to further ensure that liquid biopsies are implemented in NSCLC and become standard when tissue biopsies are absent or not possible. 

Turn-around times is another challenge, since the quick reporting of results is critical to ensuring the administration of an appropriate treatment. Delayed reporting often leads to the initiation of untargeted approaches with dismal outcomes [110] and should thus be avoided. Nevertheless, turn-around time for NGS has been reduced and is comparable to sequential approaches [86]. Importantly, novel technologies dramatically reduced turn-around times to ~24 h [111]. While novel technologies constantly reduce turn-around times in NGS, qPCR, especially highly automated assays, still offer significantly faster results, and it was demonstrated that a combination of ultra-fast qPCR for EGFR combined with subsequent NGS can be implemented, allowing both quick reports on common oncogenes and broad testing in a timely and cost-effective manner [112]. The type of NGS test is also of particular concern, with assays commonly relying on either amplicon-based [113] or hybrid-capture-based tests [114]. While both principles allow reproducible tests, the optimal solution is also dependent on the size of the panel used, with NGS panels commonly spanning a few (~10) to hundreds or thousands of genes, which can also be achieved with in-house solutions [111,115]. However, for larger panels with eg > 500 genes, validation is critical, and an outsourced approach might be more relevant, especially when demand is low. Access to clinical trials and needed research might be critical to determining the optimal size, as only few gene targets are actually associated with an approved therapy and consequently, larger panels might be more suitable in academic centers but unnecessary for routine clinical care [116]. 

Limitations to NGS still exist, and, in particular, the detection of gene fusion events using NGS lacks sensitivity compared to IHC/FISH methods, which often warrants the additional use of those methods in patients with high prevalence of those alterations [114]. Moreover, the detection of amplifications still poses challenges in NGS compared to FISH analysis [21]. Consequently, further development to refine sensitivities in these setting is critical but should not prohibit the use of NGS.

## 4. Conclusions

In the era of precision medicine, NGS appears fundamental to offer the most appropriate personalized therapy for NS-NSCLC patients and is poised to substitute for sequential testing using individual or low multiplexing gene tests. Indeed, research leading to new targeted therapies is increasing very quickly and the new international guidelines recommends the use of NGS in order to test all of the actionable alterations simultaneously, including established but also emerging targets.

## Data Availability

No new data were created or analyzed in this study. Data sharing is not applicable to this article.

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
