# Peer review of "Molecular Profiling in Non-Squamous Non-Small Cell Lung Carcinoma: Towards a Switch to Next-Generation Sequencing Reflex Testing"

_jpm, 2022, doi:10.3390/jpm12101684_

Round 1

Reviewer 1 Report

This is a well written review that addresses the currently most important biomarkers in NSCLC. It is informative for any reader interested in an update on current molecular testing of NSCLC in the clinic.

Minor comments:

a) Line 157: "The first studies of ALK expression were based on FISH....".

To me this sounds a little bit strange, because the standard FISH assays usually do not analyse gene expression.

b) Lines 216-217: "The occurrence of this mutation constitutes an important mechanism of resistance to TKIs targeting EGFR [70]."

The expression "this mutation" irritates. In the previous sentence three types of MET alterations are described (...gene amplification in 4% of cases, activating mutation and deletions of exon 14...). Which of these alterations is important in the setting of resistance to targeted therapy. I assume they are not equally frequent? It would be helpful to clarify.

gene ampli- 214 fication in 4% of cases, activating mutation and deletions of exon 14,

Author Response

We thank a lot Reviewver 1 for reviewving our article and made corrections according to these valuable comments.

a) Line 157: "The first studies of ALK expression were based on FISH....".

To me this sounds a little bit strange, because the standard FISH assays usually do not analyse gene expression.

a) Response : We thank Reviewver 1 for this important remark and absolutely agree, we have modified the corresponding sentence: Line 157-158 : "The first studies evaluating ALK rearrangements for NSCLC (...)."

b) Lines 216-217: "The occurrence of this mutation constitutes an important mechanism of resistance to TKIs targeting EGFR [70]." The expression "this mutation" irritates. In the previous sentence three types of MET alterations are described (...gene amplification in 4% of cases, activating mutation and deletions of exon 14...). Which of these alterations is important in the setting of resistance to targeted therapy. I assume they are not equally frequent? It would be helpful to clarify.

b) Response : We thank Reviewver 1 for this important remark, indeed MET amplification is the major and well documented resistance mechanism to EGFR third generation TKI. We have modified the corresponding sentence: Line 217-219 : « The occurrence of MET amplification increases to ~20% in TKI resistant EGFR mutated NSCLC, constituting an important mechanism of resistance ».

Reviewer 2 Report

The study by Nina Pujol et al investigates the most effective and useful clinical tools for non-small cell lung carcinoma molecular profile (NSCLC). Most of the actionable mutations are often detected by qPCR assays, which are fast and relatively cheap. Unfortunately, the number of actionable mutations for NSCLC is rising, especially in association with therapy resistance. This is why the authors explain why there is a shift towards NGS testing. The authors analyze the benefit and the challenges associated with NGS testing and give a picture of current European regulamentation. The paper is well-written and informative. I have only a minor comment for the authors. Among the challenges of NGS testing in clinical NSCLC, validation of a clinical MGS assay for low input DNA is crucial, especially considering that often the tissues or specimens used are limited.

Author Response

We thank a lot Reviewver 2 for reviewving our article and for his valuable comment. We made correction according to Reviewver 2 comment.

Among the challenges of NGS testing in clinical NSCLC, validation of a clinical MGS assay for low input DNA is crucial, especially considering that often the tissues or specimens used are limited.

Response : We thank Reviewver 2 for this comment, and add the according point :

Line 327-330 : « Importantly, liquid biopsies can help to guide to adjust treatment decisions under active treatment through the detection of resistance mechanisms but can also replace tissue based testing when specimen have been exhausted or are too limited [109], a situation in which the validation of NGS assays designated for low DNA input is crucial. »
